# Discomfort in Use and Physical Disturbance of FFP2 Masks in a Group of Italian Doctors, Nurses and Nursing Aides during the COVID-19 Pandemic

**Margherita Micheletti Cremasco** [1] **, Lucia Vigoroso** [2,3,*] **, Cristina Solinas** [4] **and Federica Caffaro** [2]

1 Department of Life Sciences and Systems Biology, University of Torino, Via Accademia Albertina 13, 10123 Turin, Italy; margherita.micheletti@unito.it
2 Department of Education, Roma Tre University, Via del Castro Pretorio 20, 00185 Rome, Italy; federica.caffaro@uniroma3.it
3 Institute of Sciences and Technologies for Sustainable Energy and Mobility (STEMS), National Research Council of Italy (CNR), Strada delle Cacce 73, 10135 Turin, Italy
4 Department of Psychology, University of Torino, Via Verdi 10, 10124 Turin, Italy; cristina.solinas@edu.unito.it
* Correspondence: lucia.vigoroso@uniroma3.it

**Abstract:** Face masks represent an effective COVID-19 mitigation strategy; this study investigated the quality in use of FFP2 masks in a group of 156 frontline HCWs recruited through a snowball procedure in northwest Italy. Participants filled out an online questionnaire (January 2022) on FFP2 sources of discomfort, physical disturbance at different anatomical points and suggestions for improvement. Most of the participants (69%) reported a feeling of protection and safety, but they also reported episodes of dyspnea (70%). The majority of glasses wearers (62%) reported fogging and displacement of their glasses. Humidity and heat were the main discomfort sources (39%), followed by elastic bands (32%). Physical disturbances were frequent and heavier on the ears, nose and cheekbones. Nursing aides and nurses perceived significantly more discomfort compared to doctors and nursing aides had the highest rate of physical disturbance. To address these issues, following participants' suggestions, FFP2 masks should be redesigned to be more adjustable, with different sizes and softer fabrics. The investigation pointed out criticalities in the use of FFP2 masks related to different professional roles within the overall group of HCWs and stressed the need for an FFP2 human-centered design that accounts not only for physical needs but also for workload and task variability.

**Keywords:** COVID-19; face mask; healthcare workers; occupational safety; quality in use

## 1. Introduction

In January 2020, the World Health Organization (WHO) officially declared the emergence of a novel coronavirus outbreak characterized by human-to-human transmission and airborne droplet infection as a matter of international concern regarding public health [1]. Subsequently, the global spread of this infectious disease, known as COVID-19, was designated as a pandemic [2]. To curb the transmission of the causative agent, severe acute respiratory syndrome coronavirus 2 (SARS-CoV-2), the WHO promptly issued provisional guidelines pertaining to the utilization of Personal Protective Equipment (PPE) [3].

In the scientific literature, Filtering FacePiece (FFP) masks are frequently denoted as respirators or face filters to differentiate them from surgical masks and homemade alternatives. International standards categorize respirators according to their filtration efficiency and maximum total inward leakage, establishing three protection classes (FFP1, FFP2 and FFP3), wherein the numerical values denote increasing levels of protection (i.e., 1 = low > 80%, 2 = medium > 94% and 3 = high > 99%) [4]. Various reference standards, such as EN 149-2001 for Europe, NIOSH-42CFR84 for the United States and GB2626-2006 for China, elucidate the classification, requisite performance criteria, distinctive attributes, certification methodologies and practical applications of respirators. These standards

enable cross-country comparisons of respirator models based on their stipulated physical properties and performance characteristics. Consequently, the concurrence between FFP2 (Europe EN 149-2001), N95 (United States NIOSH-42CFR84) and KN95 (China GB2626-2006) has been affirmed through conformity assessments [5].

From a safety standpoint, FFP2 masks, or their equivalents, have demonstrated the capacity to filter out 95% of particles, thereby providing optimal protection. However, the effectiveness of these masks relies heavily on correct usage and a secure fit to the individual's face. Achieving a proper seal necessitates the meticulous adaptation of the masks to the user's facial contours. The protective efficacy of respirators hinges upon their comfort and ability to accommodate the anthropometric characteristics of the head and face [6]. Inadequate comfort levels can result in improper or noncompliant mask usage, thereby increasing the risk of viral transmission [7].

Healthcare workers (HCWs) were among the first individuals mandated to wear PPE throughout their working hours, especially when attending to COVID-19 patients [8]. Notably, FFP2 masks, or their American and Chinese equivalents, emerged as the predominant PPE employed in healthcare settings and their usage became mandatory in various other contexts such as workplaces, crowded environments, public spaces and transportation, owing to the pandemic [9]. HCWs have a higher risk of infection compared with the general population [10] and exhibit a notable seroprevalence of SARS-CoV-2 antibodies [11]. The use of face masks in the context of the pandemic has demonstrated effectiveness in reducing the risk of infection among HCWs by approximately 70% [7]. Nevertheless, suboptimal mask fit, prolonged wearing and associated discomfort significantly influence the compliance of healthcare workers [12].

Regarding the comfort of FFP2 masks in the HCWs population, facial anthropometrics play a pivotal role in determining the safety and performance of masks. Although this PPE is designed to accommodate a wide range of individuals, the sizes and shapes of masks do not adequately represent the variability in facial characteristics within different user groups [13,14]. The existing literature suggests variation in facial anthropometric measurements between males and females [15], with 14 measurements significantly smaller among females [16]. While international standards for the mass production of respirators in Europe (FFP2), USA (N95) and China (KN95) are currently being upheld [4], it appears that the prevalent type of user considered during "fit-testing" primarily comprises white males, neglecting individuals with facial dimensions falling outside the central percentiles of the distribution of these measurements. As Carvalho and colleagues [17] demonstrated, male healthcare workers exhibit a significantly higher likelihood of finding well-fitting respirators and achieving successful fits on the first attempt than their female counterparts. This observation should serve as a warning, considering that women constitute approximately 67% of the global workforce in the health and care sector [18]. Failure to account for morphometric variability is one of the reasons HCWs frequently express discomfort with masks [19–21] and, more importantly, this can compromise the fit performance, protection and safety of respiratory PPE [16,22,23].

In addition to facial anthropometrics, other variables have emerged as critical factors influencing the quality of use and effectiveness of FFP2 masks among HCWs. A recent review [11] identified a correlation across multiple studies between obesity and reported mask discomfort, which subsequently increased the risk of incorrect use and infection. Furthermore, concerning weight conditions, as documented by Zhuang et al. [15], each 10 kg increase in weight significantly affects various facial measurements, except for nose length, thereby potentially exacerbating associated issues.

In addition to the issue of ensuring adequate morphometric protection, another concern arises regarding the physical discomfort that can be caused by prolonged mask usage, resulting from contact or localized pressure. Rosner [24] conducted a study revealing that extended use of FFP2 and surgical masks led to adverse physical effects such as headaches, breathing difficulties, acne, skin abrasions, rashes, impaired cognition and disturbances in vision, communication, speech and thermal balance. Furthermore, a recent review by

Silva and colleagues [25] demonstrated that masks, particularly of the FFP2 type, were the primary contributors to the occurrence of skin lesions, followed by goggles, face shields and gloves. Hu [26] demonstrated that, among healthcare workers (HCWs) who regularly utilized FFP2 masks (averaging over 12 h per day for 3.5 months), more than 95% reported some form of skin involvement, including nasal bridge ulcers associated with prolonged use of personal protective equipment (PPE). Even when worn for fewer hours, the effects persist, with symptoms reported by 47.3% of the study population when wearing masks for more than four hours [27]. Consequently, wearing a mask for more than six hours and contact with the mask material may heighten the risk of localized dermatitis and other skin diseases [28,29]. Conversely, some authors found no correlation between mask usage duration and subjective symptoms such as headaches, lightheadedness, breathlessness, facial bruising, facial irritation, mental fatigue, physical fatigue and yawning [30].

Another under-explored factor is the association between age and mask-related issues. While certain authors have observed a higher risk of dizziness, skin injuries and headaches with increasing age [31], others have reported a negative correlation between age and headaches [32]. It should be noted that most studies addressing the critical aspects of mask usage in medical contexts have encompassed diverse healthcare professionals collectively [10,33], whereas certain studies focused exclusively on surgeons [20,34] or nursing staff [35,36].

Over the past three years, due to the ongoing pandemic, there have been periods when the use of FFP2 masks was mandatory in work and social settings, alternating with phases when it was strongly recommended for indoor and even outdoor activities. Meanwhile, HCWs have consistently adhered to continuous and regulated mask usage, both nationally and internationally [37]. This heightened level of attention is unsurprising because, as early as May 2020, predictions were made that persistent or intermittent social distancing measures would likely extend until 2022, with the possibility of resurgence in contagion cases persisting until 2024 [38]. Consequently, the use of PPE, particularly among healthcare professionals, continues to garner significant focus, especially regarding enhancing tolerability and ensuring appropriate and safe usage during extended periods of wear.

*Context and Aims of the Study*

The present investigation builds on the following:

(1) HCWs are particularly exposed to the risk of infection from COVID-19 and face masks must not only have high filtration efficacy, but also fit securely and be comfortable to wear and endure for long periods [39];

(2) Italy was the first European country to experience the spread of the SARS-CoV-2 virus at the end of February 2020, with the northern regions being the most affected, also in terms of mortality [40].

The Italian healthcare system was significantly burdened and HCWs reported physical and mental exhaustion, the anguish of difficult triage decisions and the pain of losing patients and colleagues, all in addition to the infection risk [41]. In early 2023, international news spread about the possible risks due to a new pandemic wave; therefore, in Italy, there were renewed indications of attention and extensions of the mandatory use of respirators in certain contexts. As reported in the Ministerial Circular of the Ministry of Health [42], the use of respiratory protective devices is mandatory for workers, users and visitors to health and social care facilities, including hospitality and long-term care, nursing homes, hospices, rehabilitation facilities, residential facilities, as defined in art. 44 of the Decree of the President of the Council of Ministers of 12 January 2017. In accordance with the provisions of the Decree of 31 October 2022, "Urgent Measures for the Containment and Management of COVID-19 regarding the use of respiratory protective equipment" [43], these measures have been extended until 30 April 2023 and extended to clinics and doctors' offices.

Based on the previous considerations, the present study aimed to explore the quality in use of FFP2 masks in a group of Italian HCWs, with particular emphasis on discerning

variations based on the workers' roles and individual characteristics. In particular, discomfort and physical disturbance related to prolonged and continuous use of face filter masks, particularly FFP2 masks, were investigated in a sample of doctors, nurses and nursing aides, considering also the possible effects of age, BMI and time of use. The study intended to highlight critical issues which would benefit from re-design interventions of FFP2 or guidelines for their correct use, to respect human variability while assuring safety and comfort, in light of the ongoing and, perhaps, re-emerging pandemic and future similar situations.

## 2. Materials and Methods

### 2.1. Participants

Participants were recruited through a snowball procedure, starting from authors' prior personal or professional contacts in the population of interest and then sharing an online questionnaire via social media in different hospitals in northern Italy during January 2022. Inclusion criteria were being a doctor, nurse or nursing aide and continuous FFP2 use during work shifts. This resulted in a sample of 156 HCWs.

### 2.2. Instruments and Procedure

An ad hoc questionnaire was developed for the present investigation, based on previous instruments and scales. The instrument consisted of 31 questions (28 closed and 3 open answers), divided into 6 different sections. Only the 4 sections and variables of interest for the aims of the present paper are described hereafter.

Section 1: Participants' socio-demographic characteristics. In the first section, participants' socio-demographic data, including gender, age, body mass (kg), height (m) and professional role (doctor, nurse, nursing aide), were collected;

Section 2: Perceived workload and issues in FFP2 use. In the second section, based on Van Kampen et al. [44], Matusiak et al. [45] and Peres et al. [21], participants were asked to rate on a 3-point scale (1 = mild, 2 = moderate, 3 = high) the perceived level of physical workload required for their job and the number of hours of FFP2 use per day (open question). Using a short open-ended question [21], participants were then asked to describe the positive and negative aspects when using FFP2 masks. Based on Manookian et al. [35], other relevant physical problems related the to the prolonged use of FFP2 mask were also investigated: aspects related to several respiratory tract symptoms such as suffering from episodes of dyspnea (0 = no, 1 = yes) and what type of job tasks participants were performing at that moment (open-ended question), possible issues related to eyeglasses adoption when wearing FFP2 masks (open-ended question), if the elastic band adjusters were used (0 = no, 1 = yes) and the reasons why they were used (widen the elastics, tighten the elastics, other);

Section 3: Perceived discomfort. Based on Suen and colleagues [46] and Galea et al. [47], in the third section of the questionnaire, 8 items (5-point rating scale, 1 = very low, 2 = low, 3 = moderate, 4 = high and 5 = very high) investigated participants' perception of thermal discomfort (feeling of heat), thermal discomfort (sensation of humidity), pressure on the face, breathing difficulties, listening difficulties, speaking difficulties, distraction from the job and facial skin irritation;

Section 4: Perceived physical disturbance. In the fourth and last section, based on Rosner [24], we asked participants to indicate their perceived level of physical disturbance (0 = none, 1 = mild, 2 = moderate, 3 = severe) at specific points on the head (ears, nose, under the chin and cheekbones) and freely write any suggestions for FFP2 improvement (open answer).

The questionnaire was pilot-tested with a group of 8 HCWs prior being used for the present investigation. Participants gave positive feedback and small changes were made to the questionnaire to better clarify the items for the addressed sample. The questionnaire was administered online through Google Forms and the link to access it was sent through e-mails and social media. Completion required from 5 to 10 min. The questionnaire was

anonymous and the participation was voluntary. No incentives were offered to respondents. A brief written presentation of the aim of the study preceded the first section of the questionnaire, in which it was clarified that the study exclusively addressed doctors, nurses and nursing aides who wore FFP2 masks continuously during working hours. At the end of this description, there was a request to confirm the availability to participate in the study. Then, the respondent could move to the first section. The study was approved by the Bioethics Committee of the University of Turin (Protocol n. 0175984) and it was conducted in accordance with the Declaration of Helsinki for human studies of the World Medical Association [48].

### 2.3. Data Analysis

The descriptive statistics in terms of mean values and frequencies of all variables investigated were computed. A chi-square test ($\chi^2$) and a one-way analysis of variance (ANOVA) were performed to test the possible associations between the different professional roles and the perceived level of physical workload required by the job and time of FFP2 usage. Then, the two factors related to the perceived discomfort and physical disturbance were calculated with two Exploratory Factor Analyses (EFA) with no rotation. The factors were then used as the dependent variables in the subsequent analysis. Furthermore, the BMI variable was calculated from participants' self-reported data on height and body mass as $(kg)/height (m)^2$. A multivariate analysis of covariance (MANCOVA) was then performed to test the effects of professional role (factor), age, number of hours of use per day of FFP2 and BMI (covariates) on perceived discomfort and physical disturbance. Post hoc pairwise comparison tests were then performed with Bonferroni's correction. The analysis was conducted with Statistical Package for Social Science (SPSS) statistical software version 28 [49].

An interpretative content analysis was used to analyze the open-ended questions. The responses were carefully read to obtain a general impression and then categorized by two authors in recurring themes. The analysis was conducted at a general level, taking into account the manifest content of open-ended responses [50,51]. Briefly, the open-ended responses were read, interpreted by two researchers (authors of this article) and then aggregated into content areas. Because the open-ended responses were intended to provide data to contour and complement the quantitative data, they are described here in a cursory manner.

### 3. Results

### 3.1. Descriptive Statistics

Section 1: Socio-demographic characteristics. One hundred and fifty-six HCWs took part in the study. Their main socio-demographic characteristics are represented in Table 1.

**Table 1.** Participants' socio-demographic characteristics.

| Variables | Total | Professional Role | | | Age [2] | BMI [3] |
|---|---|---|---|---|---|---|
| | | Doctor | Nurse | Nursing Aide | | |
| | Frequency (%) | Frequency (%) | | | Mean (SD) | Mean (SD) |
| Male [1] | 29 (18.6) | 17 (10.9) | 9 (5.8) | 3 (1.9) | 41.4 (12.9) | 24.8 (3.2) |
| Female [1] | 127 (81.4) | 18 (11.5) | 85 (54.5) | 24 (15.4) | 43.6 (11.4) | 23.3 (4.2) |
| Total | 156 (100) | 35 (22.4) | 94 (60.3) | 27 (17.3) | 43.2 (11.6) | 23.6 (4.1) |

[1] To identify any anthropometrics-related issues, sex is used as a differentiator rather than gender [52]. [2] One-way ANOVA test showed no significant differences between the three professional roles ($p = 0.957$), [3] One-way ANOVA test showed no significant differences between the three professional roles ($p = 0.823$).

Section 2: Workload and issues in FFP2 use. Overall, the participants mainly reported a mild (54.5%) level of physical workload in their job. In detail, doctors, nurses and nursing aides declared different perceived levels of physical workload, which was mainly moderate

for both doctors and nurses (51.4 % and 58.5%, respectively) and high for the nursing aides (55.6%). The chi-square test showed a significant positive association between professional role and perceived physical workload ($\chi^2(4) = 27.075$, $p = 0.000$) (Table 2).

**Table 2.** Contingency table for professional role and perceived physical workload.

| Professional Role | Perceived Physical Workload | | |
| --- | --- | --- | --- |
| | Mild (%) | Moderate (%) | High (%) |
| doctor | 40.0 | 51.4 | 8.6 |
| nurse | 18.1 | 58.5 | 23.4 |
| nursing aide | 0 | 54.5 | 55.6 |
| Total | 19.9 | 54.5 | 25.6 |

With regard to the FFP2 usage time, more than half of the participants (58%) declared wearing it for 7 or 8 h per day, whereas 28% wore it for more than 8 h and 14% for less than 6 h. ANOVA showed no significant differences in usage time between the three professional roles ($F_{(2,151)} = 1.503$, $p = 0.226$).

When considering the open question regarding the positive aspects related to the use of FFP2, two main areas of responses emerged, which were labelled as "feeling of protection and safety" (reported by 69% of the participants) and "feeling of comfort and adherence" (reported by 19% of the participants). On these aspects, some quotations extracted from responses clarify the researchers' interpretation:

*"I feel protected, then I can have a better relationship with patients and provide better care" [#1], "I feel protected from the spread of the virus" [#2], "I feel more protected, it fit well on my face" [#3], "with this mask I can breathe better [compared with surgical mask] because it does not stay too much tight to the mouth and the nose" [#4].*

Furthermore, the remaining 3% covered some other different aspects:

*"It allows you to breathe better than with the surgical mask" [#5], "I smell less pollution and bad odours" [#6], "Compared to the surgical mask, FFP2 mask mist less the lenses of the glasses and the high protective glasses "protect against the cold" and "it has long durability" [#7].*

The main negative aspects reported dealt with three main areas or responses, as well as a general one: difficulty breathing due to heat/humidity (39%), discomfort caused by elastic bands or the presence of glasses (32%) and generic pain/discomfort (21%). Here are examples of responses transcribed related to the difficulty in breathing due to heat/humidity:

*"The mask feels muggy when I am in very warm environment for an excessive and continuous number of hours" [#8], "I have difficulty breathing, my ears hurt and it causes deep marks on my face" [#9] and "The air condenses and remains wet inside" [#10].*

Examples of quotations related to discomfort caused by elastic bands or the presence of glasses *[#11, #12]* and related to generic pain/discomfort *[#13, #14]* are reported next:

*"Sometimes discomfort behind the ears, at times fogging of the glasses" [#11], "It hurts my nose and ears, and makes it harder to breathe" [#12]; "It causes me contact dermatitis and the elastic bands are sometimes too tight" [#13], "Skin rashes and skin irritation around the chin and the nose" [#14].*

Furthermore, the remaining 3% covered some other different aspects:

*"lints cause disturbance in the nose and mouth" [#15], "cause claustrophobia and after a while it smell" [#16]. Some doctors also reported: "Elastic band bothers the ears and frequent headache by wearing PPE at the end of the work shift" [#17], and "Some type of FFP2 masks caused me dizziness and/or headache" [#18].*

The presence of moments of dyspnea during the work shift and the type of work activity being carried out at that time were investigated. Seventy percent of the participants reported having had at least one episode. Among these, three main areas of responses emerged: 45% of cases were in routine actions *[#19]*, 39% while climbing stairs, walking or talking *[#20, #21]* and 16% in moments of physical exertion and handling of heavy loads (e.g., patients) *[#22, #23]*. Example of responses to clarify the researchers' interpretation are reported here:

> *"It usually happens a few hours after I wear it on" [#19], "Walking up and down the stairs, when rushing to see a patient or when talking to someone for a long time" [#20], "It happens when I talk and move at the same time" [#21], "When performing a cardiac message" [#22] and "Carrying heavy objects up on the stairs" [#23].*

Regarding the use of glasses, 62% of the participants reported using glasses and only 5% of them declared no difficulty in wearing the mask together with the eyeglasses. The most frequent negative aspect, for 75% of people with glasses, was the fogging of the lenses *[#24]*, while others (20%) reported other types of difficulties *[#25, #26]*:

> *"the glasses are fogged with the masks because the masks are too big and lift the glasses" [#24] and "The glasses lift off my nose and distort my visual field" [#25], " . . . they tend to slip or move the mask" [#26].*

As for the use of elastic band adjusters, 93 operators out of 156 (59.6%) declared that they used them; the main reasons for use were the need to widen the elastics (53.8%) or, on the contrary, to tighten the elastics (8.6%) and to relieve ear discomfort (12.9%). The remaining percentage (24.7%) did not provide any reason.

When considering the open question regarding possible suggestions to improve FFP2 wearability and comfort in use, 25.6% of the participants suggested the adoption of longer and adjustable elastic bands *[#27]*, 15.4% suggested the adoption of softer fabrics and 5% suggested the manufacture of masks of different sizes *[#28, #29]*:

> *"More comfort for the nose, elastic bands adjustable depending on the head size" [#27], "It would be useful if more sizes were available, since many people have a smaller or a larger face" [#28], and "Produce different size models, like Small, Medium, Large, or Extra-large" [#29].*

Section 3: Perceived discomfort. Regarding the level of discomfort perception when using FFP2 masks, the items that yielded the highest score were thermal discomfort (feeling of heat) and pressure on the face. In contrast, listening difficulties and distraction from the job reported the lowest scores. Details are reported in Table 3.

**Table 3.** Mean score and standard deviation for the items used to investigate the level of discomfort perception when using FFP2 masks.

| Variable | Perception Level [1] | | | |
|---|---|---|---|---|
| | Mean | SD | Very Low and Low (%) | From Moderate to Very High (%) |
| Thermal discomfort (heat) | 2.8 | 1.0 | 39.7 | 60.3 |
| Thermal discomfort (humidity) | 3.1 | 1.1 | 28.8 | 71.2 |
| Pressure on the face | 2.9 | 1.1 | 33.3 | 66.7 |
| Breathing difficulties | 2.8 | 1.0 | 37.8 | 62.2 |
| Listening difficulties | 2.3 | 1.1 | 54.4 | 45.6 |
| Speaking difficulties | 2.7 | 1.0 | 46.7 | 53.3 |
| Distraction from the job | 1.9 | 1.0 | 75.6 | 24.4 |
| Facial skin irritation | 2.6 | 1.2 | 51.4 | 48.6 |

[1] for readers' clarity, the results were grouped.

Section 4: Perceived physical disturbance. Regarding the level of physical disturbance at specific points on the head, ears reported the highest frequency and the highest score, followed by nose, cheekbones and under the chin. Details are reported in Table 4.

**Table 4.** Mean score and standard deviation for the items used to investigate the level of perceived physical disturbance when using FFP2 masks.

| Variable | Perception Level [1] | | | |
| --- | --- | --- | --- | --- |
| | Mean | SD | None and Mild (%) | Moderate and Severe (%) |
| Perceived disturbance on ears | 2.2 | 0.8 | 17.9 | 82.1 |
| Perceived disturbance on nose | 1.3 | 0.9 | 57 | 43 |
| Perceived disturbance under the chin | 0.8 | 0.9 | 78.9 | 21.1 |
| Perceived disturbance on cheekbones | 0.9 | 0.9 | 71.2 | 28.8 |

[1] for readers' clarity, the results were grouped.

*3.2. EFA and MANCOVA*

The EFA yielded one factor for the perceived discomfort and one factor for the physical disturbance, which cumulatively explained 47.31% and 49.51% of the variance, respectively (see Table 5).

**Table 5.** Exploratory factor analysis for the two scales considered in the study.

| Items | Factor Loadings | % Explained Variance | Cronbach's $\alpha$ |
| --- | --- | --- | --- |
| | Perceived discomfort | 47.312 | 0.837 |
| Thermal discomfort (heat) | 0.699 | | |
| Thermal discomfort (humidity) | 0.651 | | |
| Pressure on the face | 0.635 | | |
| Breathing difficulties | 0.768 | | |
| Listening difficulties | 0.659 | | |
| Speaking difficulties | 0.792 | | |
| Distraction from the job | 0.604 | | |
| Facial skin irritation | 0.672 | | |
| | Physical disturbance | 49.515 | 0.660 |
| Perceived disturbance on ears | 0.623 | | |
| Perceived disturbance on nose | 0.776 | | |
| Perceived disturbance under the chin | 0.706 | | |
| Perceived disturbance on cheekbones | 0.701 | | |

Looking at Table 6, the MANCOVA analysis showed that there were significant main effects of the professional role on both perceived discomfort ($F_{(2,147)}$ = 4.21, $p$ = 0.017) and physical disturbance ($F_{(2,147)}$ = 4.42, $p$ = 0.014), whereas age, number of hours of use per day of FFP2 and BMI had no significant effects on the dependent variables. Post hoc pairwise comparison tests with Bonferroni's correction showed that perceived discomfort among nursing aides was significantly higher compared to doctors ($p$ = 0.014). No other significant differences emerged between the three groups. Similarly, physical disturbance was significantly higher among nursing aides compared to both doctors ($p$ = 0.037) and nurses ($p$ = 0.016), whereas doctors and nurses did not report significant differences in physical disturbance ($p$ = 1.000).

**Table 6.** Results for the multivariate analysis of covariance (MANCOVA).

| Source | Dependent Variable | df | Mean Square | F | *p*-Values | Observed Power |
| --- | --- | --- | --- | --- | --- | --- |
| Professional role | Perceived discomfort | 2 | 155.935 | 4.205 | 0.017 * | 0.731 |

**Table 6.** *Cont.*

| Source | Dependent Variable | df | Mean Square | F | *p*-Values | Observed Power |
|---|---|---|---|---|---|---|
| Age | | 1 | 9.965 | 0.369 | 0.605 | 0.081 |
| Number of hours of use per day of FFP2 | | 1 | 24.282 | 0.655 | 0.420 | 0.127 |
| BMI | | 1 | 14.068 | 0.379 | 0.539 | 0.094 |
| Professional role | Physical disturbance | 2 | 29.555 | 4.419 | 0.014 * | 0.754 |
| Age | | 1 | 10.015 | 1.498 | 0.223 | 0.229 |
| Number of hours of use per day of FFP2 | | 1 | 0.155 | 0.023 | 0.879 | 0.053 |
| BMI | | 1 | 6.159 | 0.921 | 0.339 | 0.159 |

Note * *p* < 0.05.

## 4. Discussion

The main goal of the present study was to investigate the quality in use of FFP masks in a group of HCWs, while also exploring factors impacting the general discomfort and physical disturbance when wearing this PPE. Data from the submitted questionnaire showed that thermal discomfort, problems caused by elastic bands or by wearing glasses and general physical annoyance were the most prevalent aspects.

Regarding perceived discomfort, thermal discomfort appeared to be a major criticality. Thirty-nine percent of our participants reported breathing difficulties while wearing the FFP2 mask, due to humidity and heat. This result is in line with that of Hunt and colleagues [53], who reported that HCWs experienced symptoms comparable to heat strain in nearly 40% of the sample (level of heat perception from moderate to extreme). Significant increases in temperature and humidity attributed to the continuous use of FFP2 masks can lead to adverse effects on productivity [54] and thermal stress is documented in several other studies among HCWs [18,19]. However, Scarano et al. [55] recorded facial skin temperature in different breathing phases with a thermal imaging camera and showed that even if the perceived discomfort is greater for FFP2 masks than surgical ones, the facial skin temperature is lower during the inhalation and expiration phases. To shed further light on this issue, in a future development of this study, it would be useful to collect both subjective and objective measurements of breathing pressure inside the mask and perceived breathing effort.

In regard to the main issues in FFP2 use, we found that the majority of the participants reported having had at least one episode of dyspnea and, among these, nearly half of the cases in routine actions. This is reflected in the literature, with dyspnea perception and breathing difficulties ranging from 36% of health care professionals [21] to 60% of intensive care unit healthcare workers [56].

We found that 53.3% of respondents had problems communicating with patients and other medical personnel, consistent with previous studies on other health professionals [19,20,57]. Considering the pivotal role played by effective communication in safety performance, this aspect should be further analyzed in future research into emergency situations and surgical performance [20].

Among the negative aspects reported by our participants, some doctors also reported headaches related to prolonged use of FFP2 masks. The relationship between prolonged use of PPE and headaches in HCWs [58,59] has already been shown in the literature, with discrepancies about its increase as wear hours rise [24,32]. This problem should not be overlooked because headaches can reduce concentration and decrease performance [60] and HCWs themselves judge their professional performance as mildly reduced by the use of PPE [59].

When considering the perceived physical disturbance in FFP2 usage, our sample most frequently reported pressure-related, with pain behind the ears (85% between moderate and severe) reporting considerably higher percentages compared to those found in the literature (e.g., 25% in [61]; 25.3% in [27]; 32.1% in [24]). Just under half of our sample

complained about moderate or severe disturbance on the nose bridge, while other studies reported percentages of just over 40% [24] and more than 80% [62]. About one third of our subjects reported physical disturbance on the cheek/cheekbones, which is less than what was found by other studies (e.g., 81.7% [62]). These divergent percentages bring into focus the importance of adapting FFP2 masks to specific face and head regions to increase the level of comfort of this PPE, considering that previous studies pointed out different levels and areas of physical disturbance among HCWs from different countries [27,33,35].

With regard to ear and nose trouble, our participants suggested some improvements to solve the issue, such as having longer/adjustable elastic bands made of softer fabrics. Even the frequent use of adapters in our sample highlights the need to adjust the elastics to avoid tension on the ears. In fact, for the nasal bridge, adding a soft thickening in correspondence with the underwire on top of the nose leads to a decrease in localized pressure. This also enables the possibility to adapt the mask to different nose shapes, with a curved underwire to provide a better adherence precisely in the region of the nose and zygomatic bone, areas where more than 70% of the critical points are found [23].

When worn, accurate adjustment/adaptation to the face is required [25], for which, given the morphometric variability of the face, additional adapters and accessories are proposed to protect specific parts of the head and face, such as protective ears straps attached to the mask elastics [25,63], paper clip or a headband to allow ear straps to rest on these items instead of the ears or a dressing on the nose bridge [24]. In relation to the specific activities of the various operators, it must be considered that the excessive sweating generated by exhausting workdays and the adaptation made by the professionals to improve sealing of the facial region can compromise the shape and adaptation of the mask to the face, contributing both to tissue damage and safety [64].

Taking more frequent breaks during working shifts has been reported as a precautionary measure and recommendation to avoid physical disturbance during the use of FFP masks [19,24,36]. However, during pandemic and emergency situations, it may become, in general, difficult to take frequent breaks and, upon removal, PPE needs to be thrown away. Then, preventive measures to prevent skin damage, such as the application of hydrocolloid or foam dressing in the pressure regions, moisturizers and emollients [25,29,34] may be adopted. However, operators often prefer not to use creams and other products despite these indications being offered (e.g., 17.7% of the participants in the study by [27]), possibly due to the concern that creams, lotions and dressings may interfere with how tightly the mask fits, decreasing protection [24]. Targeted information campaigns, highlighting the usefulness and safety in the use of these moisturizers, could be designed to tackle this issue.

Future FFP2 development needs to reflect a more diverse group of users, with particular attention on females and minority groups [16]. International organizations have also stated a need for models to be manufactured in various sizes, although there is currently no information on standardized sizes [65]. It may be necessary to develop new respirator sizing systems to respect the morphometric differences existing not only between human groups, but also due to the effects of changes in the secular trend, as pointed out, for example, by Zhuang [13].

Regarding the possible determinants of the perception of general discomfort and physical disturbances, our study showed that both aspects appear to be significantly related to HCWs' professional roles, with nursing aides and nurses suffering significantly more from discomfort compared to doctors and nursing aides reporting significantly higher physical disturbance compared to both nurses and doctors.

On the one hand, this might be due to the fact that different roles are associated with different levels of physical workload and different movements and effort, which result in more heat/sweat, thermal stress and humidity and increased perceived discomfort (as reported by Hunt [53]); on the other hand, they may also have PPE insufficiently adaptable to their morphometric characteristics and not suitable for the activities they have to perform in relation to the type of material, as it has been shown for women [52] and many ethnicities [16] that they are not adequately considered in the panels of measures

and benchmark tests for this PPE. It has to be acknowledged that, in our study, it was not possible to investigate the role played by perceived workload since the three professional roles considered significantly differed with regard to it, with nursing aides reporting a significantly higher workload. This issue could be addressed in a future development of the study by selecting groups of doctors, nurses and nursing aides with each reporting both high and low physical workloads.

Contrary to the existent literature, the time of usage did not report any significant association either with discomfort or physical disturbance. Several studies [11,25,27] indicated that the amount of PPE usage hours represents a risk factor concurring in adverse events onset. However, our data showed no significant impact of time of use in general discomfort and physical disturbances, in line with [30]. Our results may be due to the fact that the participants filled in the questionnaire at their preferred time (they did not have to be on duty while answering) and they were asked to rate the average daily amount of time they wear the mask retrospectively, instead of being asked how many hours they had the mask on while answering the questionnaire. This was a conscious choice by the researchers, not to interrupt the workers' activity in an emergency period, but it may have led to biased responses which affected the results. In a future development of the study, participants may be asked to answer questions regarding discomfort and physical disturbance at different times during their work shift (e.g., mid-shift and at the end of the shift), reporting also the actual FFP2 wearing hours and the specific tasks they were involved in, to monitor the relationship between these variables.

Contrary to previous evidence [26,66], in our study, the BMI did not show a significant association with reported discomfort and physical disturbance. It should be noted that in our sample, the BMI did not significantly differ among the three professional roles considered. This aspect should be further investigated and comprehended by involving a wider group of doctors and nursing aides, to make groups more comparable. Actually, the existing literature has reported that the prevalence of obesity among nurses is statistically significantly higher than among other healthcare professionals [67].

The evidence from the literature and from the present work stimulates future studies for the improvement of respirators to address, above all, the categories less covered by the current models of PPE, while also considering gender and ethnicity aspects. Collecting morphometric data of the face and head should use an inclusive approach to the anthropometric variability in different populations, referring also to technological devices and solutions (see Swennen et al. [68], who used mobile phones for the autonomous 3D scanning of the face and design of 3D-printed respirators) to implement adaptability and morphometric correspondence in size. With regard to the identification of the factors that cause greater disturbance and discomfort depending on the workload, always involving the most disadvantaged categories in usability [69] when performing testing starting from existing models, in order to redesign by modifying the choices of materials, thicknesses, shapes and dimensions, is recommended to meet specific needs that depend on the type of activity to be performed and movement entities. Furthermore, even though the WHO suggests wearing FFP2, or equivalent, masks for no longer than four hours [2], this PPE is worn for significantly longer. Indeed, according to Choudhury [56], the FFP2 mask can guarantee protective filtering capacity for up to a maximum of 6–8 h. Based on this, it is relevant that an FFP2 mask does not cause discomfort and disturbances in less time of use. Furthermore, an appropriate redesign of FFP2 masks can lead to a decrease in the number of masks changed every day and, consequently, reduce the amount of waste generated from their use. Thus, a good redesign can have not only a positive impact on users' performance and safety but positive effects at economic and environmental levels.

*Limitations*

Some limitations of the present investigation should be acknowledged. We referred to a non-probabilistic sampling procedure which limited the generalizability of our results. Further investigations based on a random sampling will give more generalizable results. In

addition, the time to collect responses was only one month. In future studies, this period could be extended.

In addition, our participants were mainly females, which did not allow sex-based comparisons. These data are, nonetheless, consistent with the fact that the female sex strongly prevails among nurses in Italy (76.45% of members of professional nurses' association are females [70]) and also among healthcare professionals in other countries [21,24,46].

It is also good to investigate such critical issues in a female sample, since, in the creation of the fit-panel, females are not adequately represented and even fewer females of different ethnic groups [71]. Furthermore, the fit-tests are mainly carried out on male samples and, for them, it is more frequent to have a better success rate for the qualitative fit-test [22], putting females in a disadvantaged position regarding the comfort in use of FFP2 masks and not providing them with a wide range of respirators to be able to choose the one that fits best. Ethnicity has emerged to be second to gender for impacting face size and shape characteristics [15] and minority ethnic groups continue to be under-represented in the reference fit panel of measurements [16].

## 5. Conclusions

The present study pointed out several criticalities in the use of FFP2 in a group of HCWs, highlighting specific needs and complaints for the different roles considered (nurses, nursing aides and doctors). Both general discomfort and physical disturbance were reported and claimed for more adjustable masks provided by the participants.

Workload associated with different professional roles was associated with different levels of discomfort while wearing a FFP2 mask, with nursing aides and nurses suffering significantly more from discomfort compared to doctors, and nursing aides declaring significantly higher physical disturbance compared to both nurses and doctors. This can possibly lead to behavioral non-compliance with safety rules and procedures, whereas properly fitting respiratory protective equipment is paramount, especially in healthcare practices in situations of urgency and ongoing pandemic. In preparation for future pandemics, it is imperative to identify solutions to manage the adverse effects of mask usage.

Having FFP2 masks that are suitable or adaptable not only to the different characteristics of the users but also to their professional workload and that can be worn for a long time during work activities without causing discomfort or actual injuries, represents a critical issue which deserves further investigation. This is particularly true with regard to the categories of nursing aides that most effect discomfort and disturbance during working activities. The attention should first be for healthcare professionals, but then more comfortable and safer FFP2 masks for different social environments, transports and other workplaces are of interest to the whole community, in this and other occasions of interventions necessary for the protection of the population. This is in line with a design-for-all approach to safety [72], whose main goal is for products to be designed for an "all-encompassing customer base and for a product to be made so that it can be used by the widest possible range of people" (p. 507). Then, although a lot of progress in the filtering performance was made, the comfort of this mask requires further improvement, considering that the equivalent classes of masks around the world (i.e., N95 and NK95) present minimal design variation.

**Author Contributions:** Conceptualization, M.M.C., L.V., C.S. and F.C.; methodology, M.M.C. and F.C.; formal analysis, L.V. and C.S.; investigation, M.M.C. and F.C.; data curation, F.C.; writing—original draft preparation, M.M.C., L.V. and C.S.; writing—review and editing, F.C.; supervision, F.C.; project administration, M.M.C. and F.C. All authors have read and agreed to the published version of the manuscript.

**Funding:** This research received no external funding.

**Informed Consent Statement:** Informed consent was obtained from all subjects involved in the study.

**Data Availability Statement:** Data available from the last authors upon request.

**Acknowledgments:** The authors would like to thank the anonymous healthcare professionals who participated in this survey and Giulia Bisone for her help for collecting the data. The authors would also like to thank Eugenio de Gregorio for his methodological support in the analysis of the qualitative data and Tiziana C. Callari for her participation in the event "the European night of researchers", held by the University of Torino on 24–25 September 2021, where she collaborated to raise awareness of the project and the correct use of FFP2 masks among local citizens.

**Conflicts of Interest:** The authors declare no conflict of interest.

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
