# Peer review of "Discomfort in Use and Physical Disturbance of FFP2 Masks in a Group of Italian Doctors, Nurses and Nursing Aides during the COVID-19 Pandemic"

_safety_

Round 1
Reviewer 1 Report
although the 'snowball procedure' could lead to a certain level of selection bias, the sample size is good and sufficiently heterogeneous;
the study theme is very interesting for the medical community and the study results are useful to be known.
the paper is well written and clearly exposed, my only suggetion is to present results by professional group and by sex, even just through a summary table
Author Response
Improving healthcare workers’ safety during COVID-19 pandemic: an ergonomic assessment of discomfort in use and physical disturbance of face masks in a group of Italian doctors, nurses and nursing aides
Ms. Ref. No.: Safety-2324953
Authors’ response to Reviewer 1
Comments and Suggestions for Authors
COMMENT: although the 'snowball procedure' could lead to a certain level of selection bias, the sample size is good and sufficiently heterogeneous; the study theme is very interesting for the medical community and the study results are useful to be known.
RESPONSE: We thank the Reviewer for the positive comment, we made sure to acknowledge the bias of the snowball procedure in the newly added limitations section (see page 12 lines 506-524).
COMMENT: the paper is well written and clearly exposed, my only suggestion is to present results by professional group and by sex, even just through a summary table
RESPONSE: We thank the Reviewer for the positive comment on the manuscript. Following his/her suggestion and the comments from other Reviewers, in this revision we revised Table 1 (now at page 5-6 lines 244-248) presenting participants’ characteristics by sex and professional group. With regard to the results, comparing data by sex may be misleading, since our sample was sex-imbalanced, as discussed in the newly added “Limitations” section (see page 12 lines 512-516). Results regarding perceived discomfort and physical disturbance items were presented by the professional group, as suggested (see table 1 e table 2).

Reviewer 2 Report
In the era of COVID-19 the present study has assessed the quality in use of face masks by healthcare users (specifically doctors, nurses, and nursing aides), also exploring the impact that eventually exert specific variables eg. age, BMI, and the time of wearing a mask. I believe this is an important study and I totally agree with the authors estimation that ''it highlights critical issues which would benefit from ergonomic interventions in the re-design or use of FFP2, to respect human variability while assuring safety and comfort, considering the ongoing and perhaps re-emerging pandemic and future similar situations''.
Here are some constructive comments to improve the presentation of this manuscript.
Methods
Participants (or sample design)
In this section, please provide information describing only the sample selection.
Please insert subtitles (authors are advised doing similar during the presentation of the Results and the Discussion) for each of the 4 subsections of your questionnaire.
I suggest inserting the α values for each scale of your questionnaire.
You are invited to give a proper description for each one of the scales used in the 3 subsections of your instrument e.g., how scales were constructed? Meanings of the minimum and maximum scores, Cronbach α (and comparisons with previous studies); clearly state the scale names in italic etc. in each subsection.
In the Statistics section you say: ‘’…the two variables related to the perceived physical disturbance, and the general perceived discomfort were calculated…’’, shall I assume that you mean the two scales?
Author Response
Improving healthcare workers’ safety during COVID-19 pandemic: an ergonomic assessment of discomfort in use and physical disturbance of face masks in a group of Italian doctors, nurses and nursing aides
Ms. Ref. No.: Safety-2324953
Authors’ response to Reviewer 2
Comments and Suggestions for Authors
COMMENT: In the era of COVID-19 the present study has assessed the quality in use of face masks by healthcare users (specifically doctors, nurses, and nursing aides), also exploring the impact that eventually exert specific variables eg. age, BMI, and the time of wearing a mask. I believe this is an important study and I totally agree with the authors estimation that ''it highlights critical issues which would benefit from ergonomic interventions in the re-design or use of FFP2, to respect human variability while assuring safety and comfort, considering the ongoing and perhaps re-emerging pandemic and future similar situations''.
Here are some constructive comments to improve the presentation of this manuscript.
RESPONSE: We thank the Reviewer for his/her positive comments. Following the Reviewer’s comments and suggestions, we carefully revised the manuscript to make it suitable for publication.
COMMENT: Methods. Participants (or sample design). In this section, please provide information describing only the sample selection.
RESPONSE: We thank the Reviewer for the thoughtful comment, and following it we revised the method section and included the participants’ characteristics in the results section (see page 5-6, lines 241-248). We furthermore provided more details about the sampling procedure, inclusion criteria and time-frame of sample selection (see page 4, lines 167-171).
COMMENT: Please insert subtitles (authors are advised doing similar during the presentation of the Results and the Discussion) for each of the 4 subsections of your questionnaire.
RESPONSE: Subtitles have been added to the “Instruments and procedure” and “Results” sections, as requested (see page 4 lines 177,180,193,199 and page 6 lines 242,250,328,336). Furthermore, in the “Results” section, also subtitles related to descriptive statistics vs factor and multivariate analysis have been added, for better reader’s clarity (see page 8 line 343). As far as the discussion is concerned, the results of the 4 sections have been integrated to interpret them effectively, but whenever possible, we added references to the different sections of the questionnaire as requested by the Reviewer.
COMMENT: I suggest inserting the α values for each scale of your questionnaire.
RESPONSE: We thank the Reviewer for this suggestion, the Alpha values of each computed scale is indicated in Table 5 (see page 8 lines 347).
COMMENT: You are invited to give a proper description for each one of the scales used in the 3 subsections of your instrument e.g., how scales were constructed? Meanings of the minimum and maximum scores, Cronbach α (and comparisons with previous studies); clearly state the scale names in italic etc. in each subsection.
RESPONSE: Actually, only Section 3 and 4 included proper scales, and their items underwent an EFA and a reliability analysis (Alpha coefficients are reported in Table 5). These scales’ names are now reported in italics, as requested by the Reviewer and much information has been added on the reference sources (see page 4 lines 180-200). Actually, it is not possible to compare our reliability coefficients with those of the source references since in those studies no alpha coefficients were computed: the authors referred to their items as representing one construct but then analyzed each item separately. This is also the reason why we computed the EFAs, to verify that those items can effectively represent a single factor. Items of Section 2 were single items and open questions, and they were analyzed and commented separately. Sources of items and scales have been better clarified in the “Instruments and procedure” section of this revision (see page 4 lines 180-200).
COMMENT: In the Statistics section you say: ‘’…the two variables related to the perceived physical disturbance, and the general perceived discomfort were calculated…’’, shall I assume that you mean the two scales?
RESPONSE: We thank the Reviewer for noticing this typo, actually we referred to the two factors which emerged from the EFA, which were then used as dependent variables in the subsequent MANCOVA. The text was corrected, accordingly (see p. 5 line 222).

Reviewer 3 Report
none
Author Response
Improving healthcare workers’ safety during COVID-19 pandemic: an ergonomic assessment of discomfort in use and physical disturbance of face masks in a group of Italian doctors, nurses and nursing aides
Ms. Ref. No.: Safety-2324953
Authors’ response to Reviewer 3
Open Review
( ) I would not like to sign my review report
(x) I would like to sign my review report
Quality of English Language
( ) English very difficult to understand/incomprehensible
( ) Extensive editing of English language and style required
( ) Moderate English changes required
( ) English language and style are fine/minor spell check required
(x) I am not qualified to assess the quality of English in this paper
|
|
Yes |
Can be improved |
Must be improved |
Not applicable |
|
Does the introduction provide sufficient background and include all relevant references? |
( ) |
(x) |
( ) |
( ) |
|
Are all the cited references relevant to the research? |
( ) |
(x) |
( ) |
( ) |
|
Is the research design appropriate? |
( ) |
(x) |
( ) |
( ) |
|
Are the methods adequately described? |
( ) |
(x) |
( ) |
( ) |
|
Are the results clearly presented? |
(x) |
( ) |
( ) |
( ) |
|
Are the conclusions supported by the results? |
( ) |
(x) |
( ) |
( ) |
Comments and Suggestions for Authors
COMMENT: none
RESPONSE: Following the Reviewer’s checklist, the different manuscript sections were improved.

Reviewer 4 Report
This is a useful investigation on a timely subject.
It should be revised to better convey the intended message, by improving the overall organization and reaching a balance between the narrative content and actual quantitative results.
The objectives and conclusions should match, i.e. unequivocal conclusions/highlights should be clearly stated, bringing novel information beyond what was already presented in the introduction.
When revising, please consider the following:
(A) The time window for data collection must be specified (both in the abstract and manuscript). In addition, in the manuscript more details about participants' recruitment must be provided (including criteria, sample size). Information in rows 163-165 is too scarce and vague. The information provided in the section of methods would need to be re-addressed in the sub-section of limitations.
(B) Overall, the abstract is too general. It should present more specific/concrete results. Its conclusion(s) should be not only the take-home message(s) of the manuscript, but also the synthesis of the novelty brought by this particular research.
(C) The introduction is too wordy. Authors should consider revising it for improved effectiveness in conveying the context and intended message.
(D) I would kindly recommend a revision of sub-section 1.1, so the objectives are clearly stated at the end of this subsection.
(E) Please revise and reorganize Table 1. Consider cross-tabulating the information regarding the two sexes, age, and BMI across the three professional roles. Some reported issues might be rooted in these differences, rather than doctors vs. nurses or nursing aids.
(F) You collected qualitative data, but there is little information provided on the methods you employed to analyze them (lines 224-226) and the actual results. You might consider having sub-sections within the section of results.
(G) Section 2.2 should be thoroughly revised. A table here would help synthesize the information and give credibility to the instrument you used for data collection. The rationale behind the exploratory factor analysis in the section of results is not clear. Authors might consider explaining the procedure applied to assure the face validity. Moreover, Table 4 is ambiguous (results and indicators of validity, reliability, and structure are confusingly mixed-up), so it should be thoroughly revised and reorganized.
(H) Authors mentioned MANCOVA analysis. I would kindly suggest presenting the results in a table. In addition, do not overlook the assumptions regarding data distributions for this analysis to be valid (and ANOVA in Table 1).
(I) Section of discussions should be revised. If Authors decide to have subsections within the results, then corresponding subsections might help within the discussions.
(J) Limitations should be addressed in a distinct subsection and they must be clearly stated.
Additional comments:
(a) Authors might consider a title revision, making it shorter and more specific. For example, the first part "Improving healthcare workers’ safety during COVID-19 pandemic:" conveys very little about this particular manuscript and the "ergonomic" characteristic of the assessment is not apparent.
(b) Authors should double-check a consistent use of the codes for mask types throughout the manuscript, after the equivalence was properly introduced in lines 40-47.
(c) Please double-check the correspondence between data in line 231 and those provided in Table 2.
(d) Please use decimal points in Table 3.
Author Response
Improving healthcare workers’ safety during COVID-19 pandemic: an ergonomic assessment of discomfort in use and physical disturbance of face masks in a group of Italian doctors, nurses and nursing aides
Ms. Ref. No.: Safety-2324953
Authors’ response to Reviewer 4
Comments and Suggestions for Authors
COMMENTS: This is a useful investigation on a timely subject.
It should be revised to better convey the intended message, by improving the overall organization and reaching a balance between the narrative content and actual quantitative results.
The objectives and conclusions should match, i.e. unequivocal conclusions/highlights should be clearly stated, bringing novel information beyond what was already presented in the introduction.
RESPONSE: We thank the Reviewer for the general positive comment on the manuscript. Following his/her comments, different parts of the manuscript have been revised and rewritten, to better point out the focal points and clarify the study objectives and contributions. With regard to the qualitative/quantitative results, in our questionnaire the open ended questions were used to go into further details of some aspects addressed in the rating scales and to value our participants’ perceptions on those topics, thus their results are integrated with those from the quantitative part, in line with a mixed methods approach with an embedded design (Creswell, J. W., & Clark, V. L. P. (2017). Designing and conducting mixed methods research. Sage publications.)
COMMENTS: When revising, please consider the following:
(A) The time window for data collection must be specified (both in the abstract and manuscript). In addition, in the manuscript more details about participants' recruitment must be provided (including criteria, sample size). Information in rows 163-165 is too scarce and vague. The information provided in the section of methods would need to be re-addressed in the sub-section of limitations.
RESPONSE: We thank the Reviewer for this suggestion. Following it, in this revision we added more details about participants, sample selection and inclusion criteria both in the abstract and in the method section (see page 1 lines 16-18 and page 4 lines 166-171) and further discussed this information in the newly added limitations section (see page 12 line506-511).
COMMENT: (B) Overall, the abstract is too general. It should present more specific/concrete results. Its conclusion(s) should be not only the take-home message(s) of the manuscript, but also the synthesis of the novelty brought by this particular research.
RESPONSE: We thank the Reviewer for the useful comment, and following it we revised and rewrote the abstract, reporting the specific results for each of the four questionnaire sections, while also underlining the novelty of this study (see page 1 lines 19-28).
COMMENT: (C) The introduction is too wordy. Authors should consider revising it for improved effectiveness in conveying the context and intended message.
RESPONSE: The introduction has been revised, to better point out the key concepts of the paper.
COMMENT: (D) I would kindly recommend a revision of sub-section 1.1, so the objectives are clearly stated at the end of this subsection.
RESPONSE: As recommended by the Reviewer, sub-section 1.1 has been re-organised in this revision (see page 3-4 lines 133-164).
COMMENT: (E) Please revise and reorganize Table 1. Consider cross-tabulating the information regarding the two sexes, age, and BMI across the three professional roles. Some reported issues might be rooted in these differences, rather than doctors vs. nurses or nursing aids.
RESPONSE: We thank the Reviewer for this comment. Following it and the comments from other Reviewers, in this revision we added Table 1 (at page 5-6) presenting participants’ characteristics by sex and professional group. As reported in the Note of Table xxx, preliminary analyses found no significant differences between the 3 professional roles due to age or BMI, therefore we did not further include these variable in our analyses. With regard to sex, as shown in Table 1 and discussed in the newly added “Limitations” section (see page 12 lines 506-520), our sample was imbalanced. Even though this ratio mirrors that of the actual healthcare workforce (Mangiacavalli, 2022), comparing data by sex might be misleading, therefore we did not consider this variable further.
COMMENT: (F) You collected qualitative data, but there is little information provided on the methods you employed to analyze them (lines 224-226) and the actual results. You might consider having sub-sections within the section of results.
RESPONSE: We thank the Reviewer for his/her valuable comment, giving us the possibility to further improve the qualitative data section of our study. Indeed, as requested, much information is detailed in the data analysis paragraph in the method section (see page 5 lines 232-239), while results were better re-organized (see page 6-7 lines 262-326), highlighting the helpful information provided by the participants in the open-ended questions.
COMMENT: (G) Section 2.2 should be thoroughly revised. A table here would help synthesize the information and give credibility to the instrument you used for data collection. The rationale behind the exploratory factor analysis in the section of results is not clear. Authors might consider explaining the procedure applied to assure the face validity. Moreover, Table 4 is ambiguous (results and indicators of validity, reliability, and structure are confusingly mixed-up), so it should be thoroughly revised and reorganized.
RESPONSE: Following this comment and the comment from Reviewer 2, section 2.2. has been divided into sub-sections to improve the description of our instrument and missing references for its development were provided (see page 4). Actually, the source studies for the scales used in this research did not perform any factor analyses to check whether the items represent a single construct and no alpha coefficients were computed to check for the internal consistency of the scales: the authors of these previous studies referred to their items as representing one construct but then analyzed each item separately. This is the reason why we computed the EFAs, to verify that items of section 3 and 4 of our questionnaire could effectively represent discomfort and physical disturbance, respectively. In addition, we clarified that face validity was assessed by pilot-testing the questionnaire with a group of 8 HCWs prior being used for the present investigation (see page 5 lines 203-205). Furthermore, for reader clarity table 4, now table 5 (page 8-9 line 347) was revised.
COMMENT: (H) Authors mentioned MANCOVA analysis. I would kindly suggest presenting the results in a table. In addition, do not overlook the assumptions regarding data distributions for this analysis to be valid (and ANOVA in Table 1).
RESPONSE: For better reader’s clarity, MANCOVA results were also reported in table 6 (see page 9 line 359), as requested. Furthermore, although data was not reported in the current version of the manuscript, the assumption of homogeneity of variance and the normality of dependent variables were checked before performing multivariate analysis, confirming that variables investigated are normally distributed.
COMMENT: (I) Section of discussions should be revised. If Authors decide to have subsections within the results, then corresponding subsections might help within the discussions.
RESPONSE: Considering both this comment and another one from Reviewer 2, subtitles have been added to the “Instruments and procedure” and “Results” sections (see page 4 lines 177,180,193,199 and page 6 lines 242,250,328,336). Furthermore, in the “Results” section, also subtitles related to descriptive statistics vs factor and multivariate analysis have been added, for better reader’s clarity (see page 8 line 343). As far as the discussion is concerned, the results of the 4 sections have been integrated to interpret them effectively, but whenever possible, we added references to the different sections of the questionnaire as requested by the Reviewers.
COMMENT: (J) Limitations should be addressed in a distinct subsection and they must be clearly stated.
RESPONSE: We thank the Reviewer for this comment. In this revision, we added the requested “Limitations” sub-section (see page 12 line 506).
Additional comments:
COMMENT: (a) Authors might consider a title revision, making it shorter and more specific. For example, the first part "Improving healthcare workers’ safety during COVID-19 pandemic:" conveys very little about this particular manuscript and the "ergonomic" characteristic of the assessment is not apparent.
RESPONSE: We thank the Reviewer for this comment. Following it, in this revision we shortened the title and removed the reference to ergonomics.
COMMENT:(b) Authors should double-check a consistent use of the codes for mask types throughout the manuscript, after the equivalence was properly introduced in lines 40-47.
RESPONSE: As request, reference to type FFP2 mask investigated in our study was checked and corrected throughout the revised version of the manuscript
COMMENT: (c) Please double-check the correspondence between data in line 231 and those provided in Table 2.
RESPONSE: We thank the Reviewer for noticing this data discrepancy. Following it, we checked data and corrected the value in the table (see Table 2, page 5).
COMMENT: (d) Please use decimal points in Table 3.
RESPONSE: We thank the Reviewer for noticing this inconsistency, we changed Table 3 using decimal points (see page 8).

Round 2
Reviewer 4 Report
Congratulations on your sound work to substantially improve your manuscript, which now better conveys your intended message.I have only one minor observation regarding Table 6(page 9): you should use consistent notation for significance (namely p values). Actually, I would kindly recommend replacing the "Sign." in the table header.
Author Response
Discomfort in use and physical disturbance of FFP2 masks in a group of Italian doctors, nurses and nursing aides during COVID-19 pandemic
Ms. Ref. No.: Safety-2324953
Authors’ response to Reviewer 4
Comments and Suggestions for Authors
COMMENTS: Congratulations on your sound work to substantially improve your manuscript, which now better conveys your intended message.
I have only one minor observation regarding Table 6(page 9): you should use consistent notation for significance (namely p values). Actually, I would kindly recommend replacing the "Sign." in the table header.
RESPONSE: We thank the Reviewer for the general positive comment on the manuscript. As requested, “Sign.” in header of table 6 (at page 9) was replaced with “p-values”.
